# High-Dimensional Contextual Policy Search with Unknown Context Rewards using Bayesian Optimization

**Qing Feng**[*]
Facebook
qingfeng@fb.com

**Benjamin Letham**[*]
Facebook
bletham@fb.com

**Hongzi Mao**
MIT
hongzi@mit.edu

**Eytan Bakshy**
Facebook
ebakshy@fb.com

## Abstract

Contextual policies are used in many settings to customize system parameters and actions to the specifics of a particular setting. In some real-world settings, such as randomized controlled trials or A/B tests, it may not be possible to measure policy outcomes at the level of context—we observe only aggregate rewards across a distribution of contexts. This makes policy optimization much more difficult because we must solve a high-dimensional optimization problem over the entire space of contextual policies, for which existing optimization methods are not suitable. We develop effective models that leverage the structure of the search space to enable contextual policy optimization directly from the aggregate rewards using Bayesian optimization. We use a collection of simulation studies to characterize the performance and robustness of the models, and show that our approach of inferring a low-dimensional context embedding performs best. Finally, we show successful contextual policy optimization in a real-world video bitrate policy problem.

## 1  Introduction

Contextual policies are used in a wide range of applications, such as robotics [22, 30] and computing platforms [9]. Here we consider contextual policies that are a map from a discrete *context* to a set of continuous *parameters*. For example, video streaming and real-time conferencing systems use adaptive bitrate (ABR) algorithms to balance between video quality and uninterrupted playback. The optimal policy for a particular ABR controller may depend on the network—for instance, a stream with large fluctuations in bandwidth will benefit from different ABR parameters than a stream with stable bandwidth. This motivates the use of a contextual policy where ABR parameters are personalized by context variables such as country or network type (2G, 3G, 4G, etc.). Various other systems and infrastructure applications commonly rely on tunable parameters which can benefit from contextualization. For example, optimal job scheduling and load balancing parameters may differ because of workload variations [11]. Cell tower configuration or TCP configurations can benefit from finer-grained information about environmental factors [2]. Contextual policies therefore provide an interpretable, robust approach for personalizing system parameters and improving individual-level outcomes under heterogeneous conditions.

Previous applications of contextual policy optimization (CPO) in the literature, such as those in robotics, have considered the setting where with each evaluation of the policy we observe both the reward and the context. Bayesian optimization (BO) has been successfully applied to this problem [30], using a Gaussian process (GP) to model reward as a function of both parameters and context. If the context is continuous-valued, it can be incorporated directly into the GP along with the parameters;

---

[*]Equal contribution

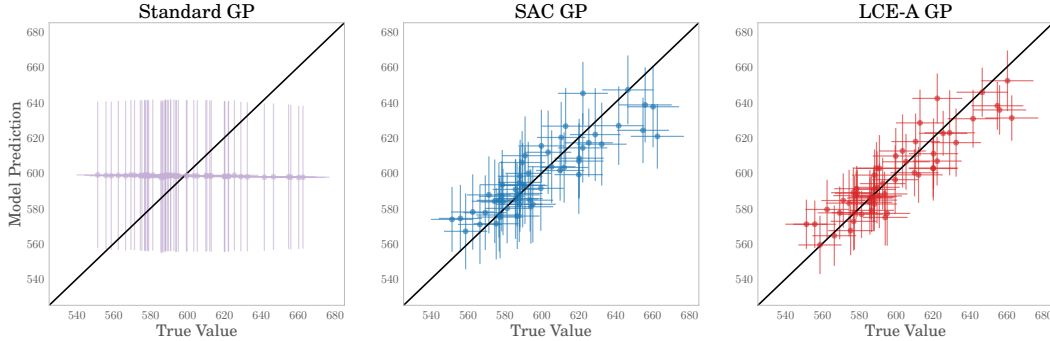

Figure 1: Leave-one-out cross validation predictions (mean, and 95% posterior predictive interval) for standard and contextual GPs fit to the results of a video playback controller experiment conducted at Facebook for a 30-dimensional policy (5 contexts and 6 parameters). The standard GP is unable to learn the high-dimensional response surface, while our proposed SAC and LCE-A models make accurate out-of-sample predictions.

if discrete, it can be included in the GP with a multi-task kernel or similar approaches for handling discrete parameters that are described below.

However, there are important CPO settings where context cannot be observed along with the rewards. In A/B testing platforms, rewards (outcomes) are measured as an aggregate across a large population that spans an entire distribution of contexts. Performing analysis by context is often not feasible, and when it is, such analyses can introduce bias by implicitly conditioning on "post-treatment" variables [31]. Without context-level data, CPO becomes a high-dimensional optimization problem where $C$ contexts each with $d$ parameters requires optimizing a function of $C \times d$ parameters.

In this work, we show that it is possible to optimize a contextual policy and get the benefits of contextualization even when rewards are measured only in aggregate. The contributions of this paper are: (1) We introduce this new, practically-important problem of contextual policy optimization with unknown context rewards. (2) We develop new GP models that take advantage of the problem structure to significantly improve over existing BO approaches. (3) We provide a thorough simulation study that shows how the models scale with factors such as the number of contexts and the population distribution of contexts, considering both aggregate rewards and fairness. (4) We introduce a new real-world problem for CPO, optimizing a contextual ABR policy, and show that our models perform best relative to a wide range of alternative approaches.

Empirically, we have found the proposed contextual GP models work well in practice for a variety of contextual policy optimization use cases, including mobile data retrieval policies, cache eviction policies, and video streaming applications. Fig. 1 gives a preview of how the methods proposed in our paper (SAC and LCE-A) enable inference and optimization of high-dimensional contextual policies using data from a real-world video streaming experiment. For reproducibility, our results in Sec. 5 leverage an open-source video streaming simulator and trace data from video playback sessions on Facebook's Android app to evaluate how these models improve CPO performance. Code for the models and replication materials are available at `https://github.com/facebookresearch/ContextualBO`.

## 1.1 Background and Related work

**Bayesian Optimization**    BO is a model-based optimization technique for settings with time-consuming function evaluations, such as A/B tests. The model is typically a GP, denoted $f \sim \mathcal{GP}(\mu(\cdot), k(\cdot, \cdot))$, where $\mu(\cdot)$ is the prior mean (usually taken as 0), and $k(\cdot, \cdot)$ is the kernel. BO is highly sample efficient, but struggles in high-dimensional settings with more than 15–20 parameters [44].

**High-dimensional BO**    Additive structure [12] has been used for handling high-dimensional search spaces in BO. [20] assumes the response surface is a sum of low-dimensional functions, each depending on only a subset of the input variables. Additional work on additive BO has developed a variety of approaches for inferring the parameter decomposition [43, 14, 42, 39, 33]. Another set

of methods for high-dimensional BO have assumed low-dimensional linear [44, 6, 36, 7, 34, 25] or nonlinear [15, 27, 32] structure to the problem.

**Multi-Task BO**  BO where observations are made for one of several tasks (such as different datasets, or here different context settings) is discussed by [40] and called multi-task BO. They used a GP with the intrinsic coregionalization model (ICM) kernel [3] to borrow strength across tasks. [18] extended the ICM kernel with a new similarity measurement to construct the inter-task covariance matrix. A similar kernel structure has been used for GPs over a mix of discrete and continuous parameters [37].

**Contextual BO**  Contextual BO is related to the much-studied contextual bandits problem [26], though with a continuous action space rather than discrete arms. [21] used BO for contextual policy search with both continuous and discrete context spaces. They used multiplicative and additive kernels to incorporate continuous context spaces into the GP, and used a multi-task GP (MTGP) for discrete contexts. The model can then be used to identify the best parameter values for any given context. [30] similarly used BO for CPO by augmenting the parameter space with the continuous context space and using a typical GP. [9] used an MTGP for CPO with discrete contexts, which enables sharing information across contexts since context rewards are often correlated. In all of this prior work, context was observed along with the reward (that is, context-level rewards) as opposed to the aggregate rewards setting we consider here.

[41] developed acquisition functions for a robotics setting with the goal of finding a contextual policy that maximizes aggregate rewards, integrated over the context distribution. Each evaluation in the policy optimization phase was of a single context, and that context was selected by the experimenter. The ability to select individual contexts for evaluation is possible in simulation or lab experiments, but not in many real-world settings, such as ours. A related line of work in robust optimization seeks to find a non-contextual policy that optimizes the aggregate rewards [45, 23], again while selecting contexts and observing rewards at the individual context level. Our work here has the same goal of maximizing aggregate rewards, but we directly observe that aggregate and cannot specify or even observe individual contexts for evaluation. Maximizing aggregate rewards using only observations of aggregate rewards allows for CPO when reward is evaluated across heterogeneous environments while avoiding the difficult credit assignment problem.

## 2 Contextual Policy Search with Multi-Task BO

In this work, we consider the case of discrete contexts. The multi-task models used in prior CPO work with discrete contexts [21, 9] assume that rewards are observed for each context, and so cannot be applied to problems with unknown context rewards. However, multi-task models form the foundation and motivation for the models that we develop for unknown context rewards. We now describe the classic MTGP here, and then develop a novel extension with latent embeddings.

### 2.1 The Intrinsic Coregionalization Model

When context $c \in \{1, \ldots, C\}$ is observed along with each reward, contextual BO can be framed as a multi-task BO problem in which each context setting corresponds to a task. The MTGP extends the GP from a single function to a collection of functions $\mathbf{f} = \{f_c(\mathbf{x})\}_{c=1}^C$ with output space $\mathbb{R}^C$ and $f_c$ the response function for context $c$. The MTGP covariance function models the covariance across both contexts and the parameter space $\mathcal{X} = \mathbb{R}^d$, that is, $k((c, \mathbf{x}), (c', \mathbf{x}')) = \text{cov}[f_c(\mathbf{x}), f_{c'}(\mathbf{x}')]$. The most common choice for $k(\cdot, \cdot)$ is the ICM kernel [8]—see [3] for a comprehensive discussion of *multi-output* GPs.

The ICM kernel assumes separability between the covariance across contexts and across parameters:

$$\text{cov}[f_c(\mathbf{x}), f_{c'}(\mathbf{x}')] = K_{c,c'} k^x(\mathbf{x}, \mathbf{x}'),$$

where $K_{c,c'}$ is a positive semi-definite (PSD) matrix that models the similarities between tasks (the task covariance matrix) and $k^x(\mathbf{x}, \mathbf{x}')$ is the kernel over the parameter space, which is shared across contexts. This is an instance of a special class of multi-output kernel functions called sum of separable (SoS) kernels. A "free-form" task covariance matrix is used in [8] written as $\mathbf{B}$, a $C \times C$ matrix, estimated using empirical Bayes. The kernel matrix for a dataset $\mathbf{X}$ can be efficiently evaluated as

$$K(\mathbf{X}, \mathbf{X}) = \mathbf{B} \otimes K^x(\mathbf{X}, \mathbf{X}).$$

The ICM kernel implicitly models each task function $f_c$ as a linear combination of independent latent functions, $f_c(\mathbf{x}) = \sum_{i=1}^{m} a_{c,i} u_i(\mathbf{x})$, where $u_i \sim \mathcal{GP}(0, k^x(\cdot, \cdot))$. The rank of $\mathbf{B}$ is equal to the number of latent functions $m$, so when $m < C$, $\mathbf{B}$ is not full rank. A low-rank task covariance matrix can induce helpful regularization [8], and also reduces the number of kernel hyperparameters from $\mathcal{O}(C^2 + d)$ to $\mathcal{O}(Cm + d)$.

## 2.2 Latent Context Embeddings

An alternative approach for handling contexts is to map them to a low-dimensional Euclidean space. This approach is especially useful when contexts take on many possible values, and is widely used in deep learning, where embeddings of discrete variables (called entity embeddings) are learned as part of the neural network during the standard training process [16].

We use this strategy to create a new kernel for contextual BO for the setting of observed context rewards. The context $c$ is one-hot encoded and passed through an embedding layer that maps it into $\tau$-dimensional Euclidean space. It is then augmented with any external or existing embeddings for the contexts. For instance in the ABR problem one might have an existing embedding of country based on network characteristics. Given an $\eta$-dimensional external embedding, each context $c$ is then represented as a point $\mathbf{z}^c \in \mathbb{R}^{(\tau + \eta)}$. The parameters of the embedding layer are jointly fit with the kernel hyperparameters by maximizing the marginal likelihood; the external embedding remains fixed. Covariance across tasks is measured with a PSD kernel function $k^z(\cdot, \cdot)$ over $\mathbb{R}^{(\tau + \eta)}$ (we use an ARD RBF kernel). Letting $E : \{1, \ldots, C\} \to \mathbb{R}^{(\tau + \eta)}$ represent the combined embeddings,

$$\mathrm{Cov}[f_c(\mathbf{x}), f_{c'}(\mathbf{x'})] = k^z(E(c), E(c')) k^x(\mathbf{x}, \mathbf{x'}). \tag{1}$$

We call this the *latent context embedding multi-output* (LCE-M) kernel.

LCE-M uses the same separability assumption as ICM, but differs in that it can incorporate external embedding information, and also differs in the way that it implicitly regularizes the function. [3] discusses the connection between GPs and frequentist kernel methods based on reproducing kernel Hilbert spaces (RKHS), and shows that the regularization of a GP prior is related to the norm of function in the corresponding RKHS. For LCE-M and ICM,

$$\|\mathbf{f}\|_K^2 = \sum_{c,c'=1}^{C} \mathbf{B}_{c,c'}^* \langle f_c, f_{c'} \rangle_k$$

where $\mathbf{B}^*$ is the pseudo-inverse of the task covariance matrix. In the ICM kernel, regularization comes from reducing the rank of $\mathbf{B}$. In the LCE-M kernel, the embedding dimension $\tau < C$ provides regularization by restricting $\mathbf{B}$ without reducing its rank, and thus without reducing the number of latent functions being used to model $\mathbf{f}$. The embedding dimensionality $\tau$ can be treated as a hyperparameter, or set with a rule of thumb like $C^{0.25}$ [1]. A similar kernel was used by [47] for categorical variables, though without the inclusion of an existing embedding for flexibly incorporating external information.

# 3 Contextual Bayesian Optimization with Aggregate Rewards

When the contexts are observed along with the rewards, the contextual policy is constructed by jointly modeling context and parameters. For the ICM kernel, the dimensionality of the model space is only $d$ with $C$ tasks; for LCE-M it is $d + \tau + \eta$. In the aggregate reward setting, we observe a single reward for the entire $(C \times d)$-parameter contextual policy, and have no choice except to solve a high-dimensional optimization problem. We develop two kernels that allow for effective BO in this space by taking advantage of the particular structure of the aggregated CPO problem. This approach fits into the broader framework of grey-box optimization by exploiting the additive structure of the objective function to improve optimization efficiency [4].

## 3.1 Structural Assumptions for Aggregated Contextual Policy Search

We consider three assumptions that enable modeling in the high-dimensional contextual policy space. We denote the full policy as $\bar{\mathbf{x}} \in \mathbb{R}^{C \times d}$, and let $\mathbf{x}_c \in \mathbb{R}^d$ represent the parameters for context $c$.

*Assumption* A1. *No interference*: A policy $\bar{\mathbf{x}}$ affects the rewards of individual units with context $c$ only via $\mathbf{x}_c$.

*Assumption* A2. *Single spatial kernel*: The (unobserved) response surfaces for each context $c = 1, \ldots, C$ have the same covariance function over the parameters $k^x(\cdot, \cdot)$. In essence, the smoothness of the response surface with respect to the parameters is the same under each context (note that how reward depends on the parameters need not be the same for each context).

*Assumption* A3. *Stable context distribution*: The distribution of context values is constant. This is required for consistency of the aggregate reward both in time and parameter space.

A1 and A2 are assumptions also made by the multi-task models in Sec. 2: A1 is implied by the kernel separability, and A2 is used when giving the same $k^x(\cdot, \cdot)$ to each task. In A3 we assume the context distribution is stable, but we do not assume it is known, and will infer it in our models here. Under these assumptions, we can derive an appropriate GP kernel for the aggregate reward response. Let $f : \mathbb{R}^{C \times d} \to \mathbb{R}$ be the aggregate reward response, $f_c : \mathbb{R}^d \to \mathbb{R}$ the unobserved unit-level reward for $c$, and $w_c$ the population frequency of $c$. Then,

$$f(\bar{\mathbf{x}}) = \sum_{c=1}^{C} w_c f_c(\mathbf{x}_c). \tag{2}$$

## 3.2 The Structural Additive Contextual Kernel

If we take $f_c$ as i.i.d. $\mathcal{GP}(0, k^x(\cdot, \cdot))$ in (2), we can derive the corresponding kernel which we call the *structural additive contextual* (SAC) kernel:

$$\text{Cov}[f(\bar{\mathbf{x}}), f(\bar{\mathbf{x}}')] = \sum_{c=1}^{C} w_c^2 k^x(\mathbf{x}_c, \mathbf{x}'_c).$$

Fitting a typical ARD kernel to the high-dimensional function would require $C \times d + 1$ hyperparameters ($C \times d$ lengthscales and an output scale). For ARD $k^x(\cdot, \cdot)$, the SAC kernel has $C + d$ hyperparameters ($d$ lengthscales and $C$ context weights $w_c$) which allows incorporating a larger number of contexts.

The SAC kernel is similar to the additive kernels used for generic high-dimensional BO described in Sec. 1.1, such as Add-GP-UCB [20], with a key difference: since those do not have the explicit $C \times d$ block structure that comes with contextualization, they use isotropic kernels across each component. The SAC kernel is able to use an ARD kernel for each context, and share lengthscales for $x_{c,i}$ and $x_{c',i}$ to better borrow strength across contexts. In that regard, the SAC kernel has a similar flavor to the ICM kernel: it is equivalent to summing the outputs of an ICM model with a diagonal task covariance matrix. In practice, the shared kernel is a significant source of ICM generalization across tasks [24], and the SAC kernel brings that same power to contextual BO.

## 3.3 Latent Context Embeddings with Aggregate Rewards

Just as the SAC kernel extends the ICM structure to the aggregate rewards setting, we can extend the LCE-M model to aggregate rewards to get the benefit of context embeddings. We give the latent context rewards $f_1, \ldots, f_C$ a joint GP prior with the LCE-M kernel. Then, combining (1) and (2),

$$\text{Cov}\left[f(\bar{\mathbf{x}}), f(\bar{\mathbf{x}}')\right] = \text{Cov}\left[\sum_{c=1}^{C} w_c f_c(\mathbf{x}_c), \sum_{c=1}^{C} w_c f_c(\mathbf{x}'_c)\right]$$

$$= \sum_{c=1}^{C} \sum_{c'=1}^{C} w_c w_{c'} k^z(E(c), E(c')) k^x(\mathbf{x}_c, \mathbf{x}'_c). \tag{3}$$

We call this the *latent context embedding additive* (LCE-A) kernel. This kernel has $(1 + \tau) \times C + d + \tau + \eta$ hyperparameters. Although it has more hyperparameters than the SAC kernel, the model complexity is not necessarily higher as the kernel $k^z(E(c), E(c'))$ imposes regularization and encourages correlations across component functions. With stronger correlation, there is a larger norm and more regularization, which results in a lower model complexity.

LCE-A maintains the benefits of the SAC kernel sharing a spatial kernel across contexts, while also explicitly modeling and accounting for the correlation in the latent context rewards. As we will see in Sec. 4, this leads to benefits in both average performance and in robustness.

# 4 Numerical Experiments

We use a series of synthetic problems to study the performance characteristics of our proposed models, including their performance relative to the observed context rewards setting; their ability to scale to large numbers of contexts; their ability to improve outcomes for individual contexts, including rare contexts; and their performance under model misspecification.

The results here focus on a family of synthetic benchmarks based on the Hartmann6 test function. The parameter space is taken to be the first five dimensions of the function ($d = 5$), and the last dimension is used as an unobserved latent context dimension. The latent context values are spaced uniformly across the sixth input dimension. We then compute aggregate rewards with the formula in (2). This benchmark construction allows us to assess BO performance on CPO with aggregate rewards while varying both the number of contexts $C$ and the weight distribution $w_c$. Because we can compute the true latent $f_c$, we can also compare with multi-task models that have access to context-level rewards. For BO, we use EI [19] as the acquisition function, though our models are agnostic to the acquisition function. In all benchmarks, we consider at least four methods: quasi-random search (Sobol) [35], BO with a standard ARD Matérn 5/2 kernel [38] (Standard BO), and BO with the SAC and the LCE-A models. We used embedding dimension $\tau = 1$ for LCE-A. Results with alternative synthetic functions are given in the Appendix and are qualitatively the same. All plots show the mean and 95% confidence intervals (two standard errors) across 15 runs.

**Scaling with number of contexts.**   We first consider the setting where each context has equal weight $w_c = \frac{1}{C}$. Fig. 2 shows the results of using BO for CPO with aggregate rewards (StandardGP, SAC, and LCE-A), as well as with fully observed context rewards (ICM and LCE-M). In the latter case, BO was performed separately per context. With aggregate rewards, the dimensionality of the search space grows linearly with the number of contexts $C$. The performance with the standard GP rapidly degraded while the other methods found significantly better policies across the full range of $C$. Because the LCE-A is able to borrow strength across contexts, it scales better with the number of contexts relative to the other aggregate reward methods, as indicated by a slower decrease in final reward as $C$ grew.

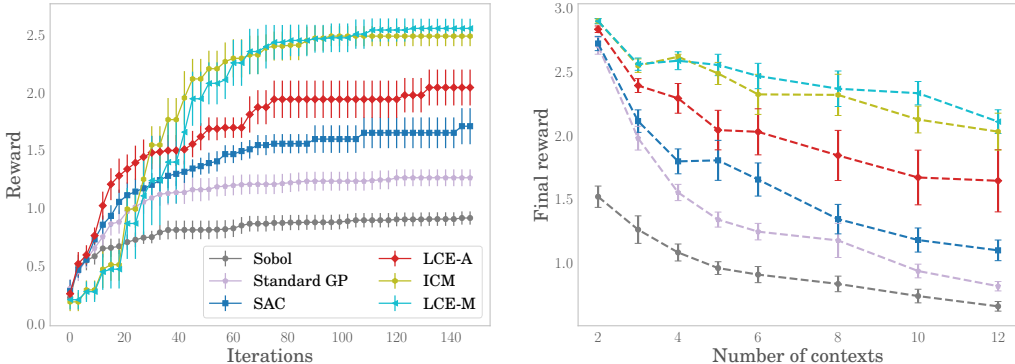

Figure 2: Contexual Hartmann5D with uniform weights. (Left) benchmark traces (average best found across 15 runs, with 95% confidence interval over 150 trials) for each method for $C = 5$ contexts with equal weight. LCE-A performed best among methods using aggregated reward. The gap between LCE-A and LCE-M indicates the value of knowing context-level rewards. (Right) Final reward of 200 trials for different numbers of contexts. The best value found by LCE-A decreased much slower with more contexts, showing the scalability of high-dimensional contextual policy search.

Fig. 2 also demonstrates that observing context-level rewards (ICM and LCE-M) unsurprisingly improves BO performance. However, aggregate-rewards CPO with LCE-A was able to capture much of the benefit of CPO. Furthermore, for $C \geq 5$ the regret gap between LCE-A and multi-task models was constant: aggregate-rewards CPO did not scale worse with $C$ than multi-task CBO. This result shows that, with LCE-A, CPO can be a valuable tool in settings where observing context-level rewards is infeasible.

**Robustness to the context distribution.** In many applications some contexts make up only a small portion of the total population, which can make it especially challenging to optimize policies in the aggregate reward setting. We assessed CPO performance in this setting by skewing the weight distribution: a small number of contexts were given the majority of the weight (called *dense contexts*), and the remaining (*sparse contexts*) were each given weight $w_c = 0.01$ (where weights sum to 1).

Fig. 3 shows the optimization performance on this problem, for two and five dense contexts, while varying the total number of contexts. LCE-A remained the best-performing method in all scenarios. With a skewed context distribution, the effective dimensionality of the problem is lowered because optimizing for dense contexts alone will capture most of the available reward. Because of this, the best reward found by both the SAC and LCE-A models remained essentially constant when increasing the number of sparse contexts.

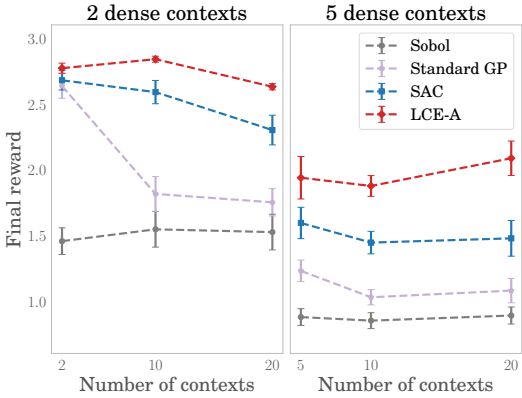

Figure 3: Best reward found for contextual Hartmann5D with a skewed context weight distribution over 100 trials. $s$ sparse contexts were each given weight 0.01, and dense contexts split the remaining $1 - 0.01s$ weight. Increasing the number of sparse contexts did not significantly alter the optimization.

One potential concern with optimizing contextual policies without observing context-level rewards is that the policy for rare contexts could be poorly identified and the contextual policy could degrade their experience. This relates to the issue of fairness in machine learning [10]. Fig. 4 shows optimization traces for one of the sparse-context problems of Fig. 3 in which aggregate reward was optimized with 5 dense and 5 sparse contexts (total weights of 0.95 and 0.05, respectively). The left figure shows performance on the aggregate reward across all 10 contexts (the objective of the optimization), while the right figure shows the reward of just the sparse contexts.

The SAC model did well at optimizing the top-level aggregate reward, but performance on the rare contexts was no better than random. Since the SAC model assumes independence across contexts, the model is unable to infer the latent reward responses for the rare contexts and cannot craft a good policy. The LCE-A model, on the other hand, is able to borrow strength from the dense contexts to the sparse contexts, and performed well even on the sparse contexts. This result shows that, with LCE-A, CPO with aggregated rewards is robust to rare contexts and can produce policies that improve performance for all contexts.

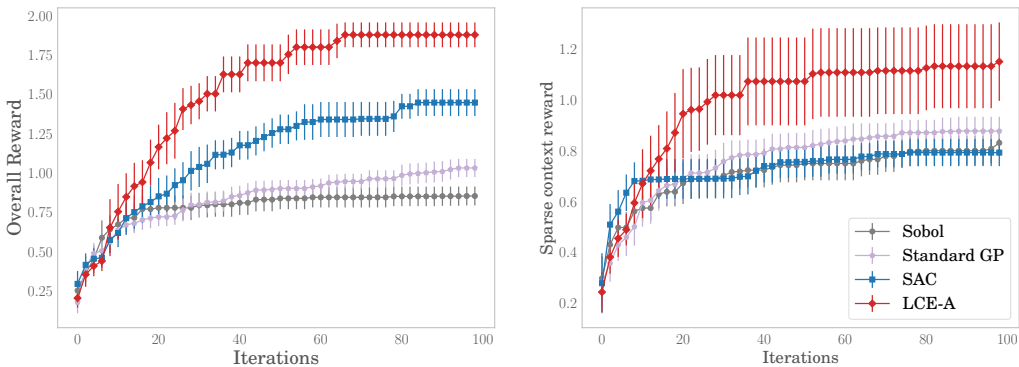

Figure 4: Optimization performance with 5 dense and 5 sparse contexts. (Left) Overall optimization performance over 100 trials. (Right) Performance on the sparse contexts. Compared with other models, LCE-A produced the most fair results and significantly improved the policy for the rare contexts.

**Dynamic context distributions.** In real-world settings like internet services or mobile applications, policies may cause individuals to change their context. For example, a poor experience with video conferencing over a cellular network may cause individuals to change to wifi when possible. This means context weights may depend on the unobserved context-level rewards, and thus depend on the policy parameters in direct violation of Assumption A3.

To study the impact of this on our models, we consider the case in which the context distribution shifts according to the rewards. Specifically, we set $w_c(\bar{\mathbf{x}}) = f_c(\mathbf{x}_c) / \sum_{c'} f_{c'}(\mathbf{x}_{c'})$, so that weight shifts towards contexts with higher unit-level rewards. Fig. 5 shows the optimization performance with $C = 50$ contexts. Despite the violation of A3, both the SAC and LCE-A models were able to effectively optimize and found policies much better than random and standard BO.

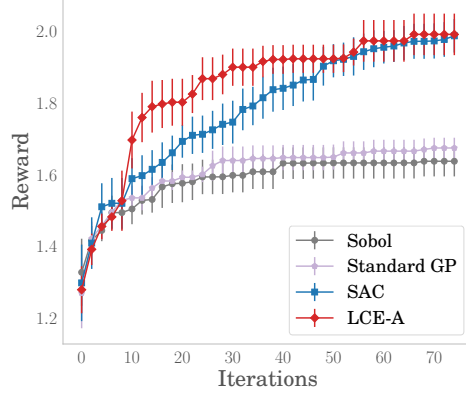

Figure 5: Optimization with 50 contexts and a dynamic context distribution. LCE-A performed well despite the violation of Assumption A3.

## 5 Contextual Policy Search in a Realistic Environment

We now study contextual policy search in a realistic environment: ABR video playback policies in heterogeneous mobile environments, using de-identified trace data from the Facebook Android mobile app. ABR algorithms determine the bitrate of each video chunk (here a one second segment of video) based on observations from the network (e.g., bandwidth fluctuation) and the video stream (e.g., playhead location). The goal is to maximize the quality of experience (QoE). Following the literature on model predictive control for video streaming [46], we consider an operationalization of QoE that is a weighted sum of the quality of the video chunks, stall time, and inter-temporal variation in quality. With this quality function, the goal of the playback controller is to request the highest quality chunks while ensuring an experience that is free of stalls.

We consider a contextual ABR policy that varies the parameters of an ABR controller based on the device's current *connection quality*. We evaluate our method with a modified video streaming simulator based on the *Park* platform [29]. The simulator mimics real-world network conditions based on de-identifed measurements from video playback sessions on Facebook's mobile android client. This allows us to consider the connection quality, fluctuations in available bandwidth, latency, and compression rates for each playback session.

The contextual policy is specified as a 48-dimensional vector, which specifies four of parameters of a linear controller for 12 possible contexts (see Appendix S1.1 for details). Playback sessions were classified using a 12-category clustering of connection quality based on initial measurements of throughput, latency, and connection type (cell or WiFi).

Fig. 6 shows optimization performance on the task of finding a contextual policy that maximizes QoE. In addition to the methods in Sec. 4, we compared a broad collection of existing methods: additive kernel methods Add-GP-UCB [20] and Ensemble BO (EBO) [43]; linear subspace BO methods ALEBO [25], HeSBO [34], and REMBO [44]; trust-region method TuRBO [13]; and the evolutionary strategy CMA-ES [17]. See Appendix S1.3 for the details for each comparison method. We also compared to doing regular BO on the 4-d non-contextual policy, that uses the same controller for all 12 contexts. All BO benchmarks were initialized with 8 quasi-random (Sobol) points. The benchmark results were averaged over 25 runs.

Fig. 6 (left) shows that the LCE-A method learned the best ABR controller policy. Sobol and Standard GP failed to handle the 48-dimensional search space. The linear embedding methods performed better than random, however there is no linear low-dimensional structure to this parameter space, so their final best QoE was 25% worse then LCE-A. Add-GP-UCB and TuRBO had better performance, and achieved QoE 6% less than LCE-A. By leveraging the problem structure and sharing information among contexts, LCE-A and SAC demonstrated higher efficiency than Add-GP-UCB. We excluded CMA-ES from the figure because it was unable to achieve a QoE better than the non-contextual policy

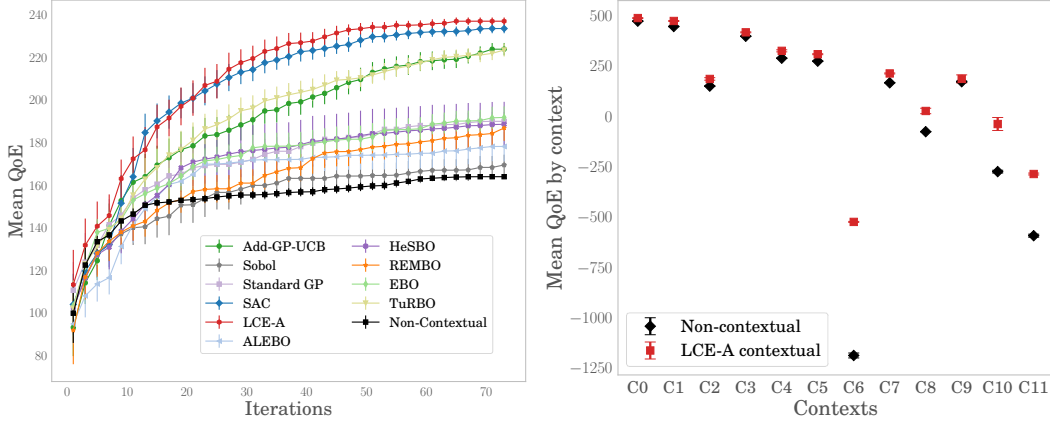

Figure 6: (Left) Average optimization performance for ABR policies with 4 parameters and 12 network connection contexts (error bars show two standard errors of the mean). LCE-A produced the contextual policies with the highest rewards (Right). LCE-A was able to improve the policy for each context (connection quality cluster) relative to the non-contextual policy. Contexts are sorted by mean latency of the cluster, and show that contextualization provided significant benefit for devices with high latency like C11

(i.e $150 \pm 10$). There was significant benefit to contextualization, with LCE-A producing a 50% increase in QoE relative to the non-contextual policy. Importantly, LCE-A also produced significant improvements for each context individually (Fig. 6, right). For example, the context C6 had the worst QoE under the non-contextual policy. This context had low bandwidth and produced only 1.8% of the total trace population, so it made only a small contribution to the aggregate reward and was not well-supported by the non-contextual optimization. LCE-A was able to significantly improve the outcomes in this context, despite its small weight. Even with aggregate rewards, contextual policy search with LCE-A is able to improve outcomes for all contexts. Besides the optimization performance, results from contextual BO method are also highly interpretable. See Appendix S1.2 for more discussion and visualizations.

## 6 Conclusion

We have shown that it is possible to deploy and optimize contextual policies even when rewards cannot be measured at the level of context. This enables the use of contextual BO in a broad range of applications where rewards are measured only in aggregate, as is common in A/B testing platforms. The LCE-A model makes it possible to optimize in high-dimensional policy spaces by leveraging plausible inductive biases for contextual policies. This not only improves top-level aggregate rewards relative to non-contextual policies, but also improves the fairness of the policy by improving outcomes across all contexts. The LCE-A model in particular is robust to the presence of rare contexts and can improve outcomes for small populations.

Our work aims to highlight a new, empirically-motivated problem and a class of structured solutions to this problem. There are many plausible extensions to this work, including more flexible models that include hierarchical priors for lengthscales across tasks. For instance, we can relax Assumption A2 by extending to the linear model of coregionalization (LMC) and determine model complexity adaptively. We consider a data regime under which few policies can be evaluated, but we expect that more function evaluations could enable modeling of population shifts caused by the policy (e.g., $w_c$ is a function of $\bar{x}$). Finally, we hope that future work can consider leveraging pre-trained, unsupervised representations of contexts to reduce the burden of learning good embeddings of contexts from scratch, which would further enable our method to scale to a very large number of contexts.

## Broader Impact

The methods introduced in this paper expand the scope of problems to which contextual Bayesian optimization can be applied, and are especially important for settings where policies are evaluated with A/B tests. We expect this work to be directly beneficial in this setting, for instance for improving services at Internet companies as in the ABR example that we described in the paper. We are including our complete code for all of the models introduced in this paper, so the work will be immediately useful. As shown in the paper, contextualization improves not only the top-line performance of policies, but also improves the fairness of policies by improving outcomes specifically for small populations that do not achieve good performance under an existing non-contextual policy. This work will directly benefit these currently under-served populations.

## Acknowledgements

We thank Denise Noyes for her guidance on the ABR application and help with data, and Shaun Singh for his help with deploying the live experiments which validated the value of the methods proposed here. We thank David Arbour for his feedback on the paper.

## Funding Disclosure

There is no funding in direct support of this work.

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
