[Supplementary Material]

# Supplemental Materials: High-Dimensional Contextual Policy Search with Unknown Context Rewards using Bayesian Optimization

## S1 Additional Experiment Results

### S1.1 Video Playback Experiment

In our experiment (Sec. 5), the ABR controller determines the video bitrate as a function of the current playback buffer and past network observations. Specifically, it computes a bitrate value $v_t$ for each video chunk $t$ with

$$v_t = p_b \tilde{b}_t + p_o o_t + p_c, \tag{S1}$$

where $\tilde{b}_t$ is the bandwidth estimation to download chunk $t$, $o_t$ is the playback buffer occupancy, and $p_b$, $p_o$ and $p_c$ are parameters of the model for BO to optimize. A bandwidth estimator computes $\tilde{b}_t$ with an exponential average over the past observations:

$$\tilde{b}_t = \sum_{t'=0}^{t} \exp\left[-p_w(\tau_t - \tau_{t'})\right] b_{t'} / \sum_{t'=0}^{t} \exp\left[-p_w(\tau_t - \tau_{t'})\right], \tag{S2}$$

where $b_{t'}$ is the bandwidth measurement at $t'$, $\tau_t$ is the wall time when chunk $t$ is downloaded and $p_w$ is a weighting parameter for the exponential average.

With the value $v_t$, the controller selects the actual bitrate with a thresholding method similar to [28]. Specifically, it picks the maximum bitrate index $i$ such that its corresponding threshold $h_i$ is below the value $v_t$. Each threshold level is computed with a linear function $h_i = p_s + p_h i$, where $p_s$ and $p_h$ are set to be 1 and 3.

In total, the controller contains 4 parameters to search over, per context. The specific range of the parameters are in Table S1. In our simulator, we adapted the Park framework [29] to utilize 1912 traces of video playback sessions from the Facebook Android app, and estimated chunk sizes for each chunk of each playback session. Additional latency was added to chunk requests using context-level means from pre-defined connection quality clusters. This setup allows us to capture real-world heterogeneity in available bandwidth, as well as any differences in compression rates that may be correlated with network latency. Traces were classified according to a clustering of connection characteristics. Summary statistics about these clusters are given in Table S2. Clusters are sorted and labeled according to their mean latency for readability. One can see from the table that clusters vary in size, vary by an order of magnitude in mean bandwidth (BW), and vary significantly by latency (latency tends to be higher on clusters that include a higher proportion of connections on cell networks).

| $p_b$ | $p_o$ | $p_c$ | $p_w$ |
|---|---|---|---|
| $[0, 1]$ | $[0, 3]$ | $[0, 1]$ | $[0.0001, 0.25]$ |

Table S1: Parameter ranges.

### S1.2 Interpretation of the Contextual ABR policy

The LCE-A model is a grey-box approach to Bayesian Optimization, thus yielding highly interpretable results. We first examine how the learned context embeddings relate to the mean network bandwidth and latency of the corresponding context cluster. In Fig. S1 (left), each dot corresponds to each context, colored by the learned 1-d embedding values (Similar embeddings are collapsed to use the same color). The learned embeddings roughly capture the context heterogeneities in throughput and latency. This yields different optimal policies for each context. Fig. S1 (right) shows the change of objective metric QoE with respect to the parameter $p_b$ that controls bitrate change based on the network bandwidth. Contexts with low-throughput, colored in blue, respond more slowly to bandwidth changes.

We also visualize a 2-d slice of LCE-A model predictions with respect to parameter $p_b$ and $p_o$ for context C0 and C11. The two contexts have very different network characteristics: context C0 has

| Context | % Traces | On wifi (%) | Mean latency (ms) | Mean BW (Mbps) | Median BW (Mbps) |
|---------|----------|-------------|-------------------|----------------|------------------|
| C0  | 10.7 | 100.0 | 47  | 3.09 | 2.85 |
| C1  | 15.6 | 100.0 | 58  | 2.75 | 2.37 |
| C2  | 11.9 | 100.0 | 59  | 0.96 | 0.85 |
| C3  | 4.0  | 80.8  | 75  | 2.48 | 2.05 |
| C4  | 5.1  | 40.8  | 82  | 1.42 | 1.02 |
| C5  | 18.8 | 0.0   | 91  | 1.83 | 1.56 |
| C6  | 1.8  | 84.4  | 92  | 0.20 | 0.18 |
| C7  | 11.6 | 100.0 | 96  | 1.03 | 0.79 |
| C8  | 7.2  | 26.4  | 129 | 0.66 | 0.43 |
| C9  | 3.1  | 84.7  | 139 | 1.81 | 1.31 |
| C10 | 2.9  | 66.5  | 157 | 0.63 | 0.29 |
| C11 | 7.3  | 63.1  | 210 | 0.39 | 0.26 |

Table S2: Summary statistics of *connection quality clusters* used as context variables in the ABR simulation.

Figure S1: (Left). Scatter plot of the log of context mean network latency and bandwidth. Contexts are labeled and colored based on the learned embedding values. Contexts with similar embeddings have similar network latency and throughput. (Right). A 1-d slice of LCE-A predictions for each context, showing the response of the objective metric with respect to the parameter controlling bitrate change.

high network bandwidth and low latency, while C11 has low bandwidth and high latency. Fig. S2 shows very different policy patterns and optimums for the two contexts (darker red means higher QoE). It can be seen that contexts with better network conditions benefit from having higher values in parameter $p_b$ and $p_o$ that could be more aggressive in increasing video quality.

### S1.3   Method Implementations

The LCE-A and SAC models were implemented using BoTorch, a framework for BO in PyTorch [5]. The linear embedding methods (REMBO, HeSBO, and ALEBO) used the implementation of [25], and a linear subspace dimension of $d_e = 8$ for all three methods. The methods Add-GP-UCB, TuRBO, EBO, CMA-ES used reference implementations from their authors with default settings. See the reproduction code at `https://github.com/facebookresearch/ContextualBO` for the exact calls used for each method.

### S1.4   Additional numerical experimental results

**Independent latent context functions** In this simulation, we consider the case where latent context functions are independent with each other, so there is no value to sharing information across contexts. The parameter space was 5-dimensional and the latent functions $f_c$ were drawn from the same GP prior with zero mean function and an ARD Matérn 5/2 kernel. The outputscale was 1.0 and, lengthscales ranged from 0.25 to 0.5, and there were 8 contexts with equal weight. Fig. S3 shows the

C0: high throughput, low latency      C11: low throughput, high latency

Figure S2: A slice of LCE-A predictions for two context values and two ABR controller parameters. The context C0 and C11 have very different network characteristics, thus very different policy patterns. The optimal policy for C0 prefers higher values in parameter $p_b$ and $p_o$, while for C11 it is the opposite.

benchmark traces. Both SAC and LCE-A obtained the best values with a not-statistically-significant difference. This shows that LCE-A can continue to perform as well as SAC even when there is no correlation across contexts. The result also indicates that the performance gap between SAC and LCE-A decreases as context correlation goes to zero.

Figure S3: Benchmark trace (average best found across 10 runs, with 95% confidence interval) across methods for $C = 8$ contexts with equal weight. Each latent function was independently sampled from the same GP prior. SAC and LCE-A reached a similar final reward. Due to the independence across contexts, SAC converged to the optimum faster.

**Dynamic context weights**     With the same generative model described in Sec. 4, we evaluated the methods across 10 and 20 contexts. As in Fig. 5, the LCE-A method performed well even with the violation of Assumption A3.

**Branin2D function**     In these simulations we used as the latent functions the 2-dimensional Branin function, scaled by an exponential function $\exp(-u_c \exp(\frac{5+x_{c0}+x_{c1}}{30}))$, in which $u_c$ is the 1-d dimension context embedding spaced evenly between $[0, 1]$ and $\mathbf{x_c}$ is the 2-d parameter vector. In Fig. S5, we show the results of optimization evaluations with 5, 10, and 15 contexts with equal context weights. The LCE-A models consistently performed best. As the number of contexts increased, the gaps in final reward between LCE-A and other methods grew.

Fig. S6 uses a skewed weight distribution with 5 dense contexts and 5 sparse contexts (each with weight 0.01, as in Sec. 4). The left plot shows the benchmark traces of aggregate rewards. LCE-A

Figure S4: Contextual Hartmann3D with dynamic weights. Benchmark trace (average best found across 15 runs, with 95% confidence interval) across methods for 10 and 20 contexts.

Figure S5: Contextual Branin2D for 5, 10 and 15 contexts with equal weight. Benchmark traces (average best found across 15 runs) show LCE-A outperformed other methods in all three cases.

obtained a significantly better final reward than other methods. Since the dimensionality of the parameter space is low, standard BO method performed reasonably well at optimizing aggregate reward. The right trace plot shows the reward just for sparse contexts. LCE-A performed well for these rare contexts while SAC and standard GP were similar to random search. This again highlights the robustness of LCE-A in handling rare contexts, besides improving overall reward.

Figure S6: Contextual Branin2D for 10 contexts with a skewed weight distribution (5 dense contexts and 5 sparse contexts). The left plot shows average best found across 15 runs. LCE-A found the best aggregate reward. The right plot shows the trace for sparse contexts only. The LCE-A method performed robustly in optimizing rewards even of the sparse contexts.