[Reviews · NeurIPS 2020]

Review 1

Summary and Contributions: This paper addresses the problem where the underlying generative process has a context component but the observation cannot be made for separated context. Since the observed reward is an aggregation of context-wise rewards across different contexts, the paper proposes two different kernels which have additional modeling capability to explain the contributions of the reward of each context. SAC assumes independence among context-wise rewards and LCE-A models dependencies between contexts via en embedding.

Strengths: - The paper links a simple tweak of existing kernels and the interesting problem where the function varies depending on contexts, but context-wise function values are not directly observable. - The characteristic of the ICM kernel using a single spatial kernel is well observed and effectively utilized in aggregated reward CPO. - By sharing parameters of the kernel for context-wise reward function, this approach has less number of parameters compared to the naive approach. (This is clear for SAC, but the analysis is not performed for LCE-A. Can you also provide this?) - In the ablation study, various aspects of the method are investigated.

Weaknesses: - Even though it is motivated from the ICM kernel, Assumption 2, sharing a spatial kernel, relies on the quite strong assumption that policy-reward behaviors are quite the same across contexts. On the other hand, in experiment results, it seems that disadvantages from such strong assumption are recovered by the advantages from using data from all contexts to train the shared spatial kernel parameters, which may allow kernel parameters to receive more training signal and to avoid overfitting. Therefore, checking how robust LCE-A is when policy-reward behavior changes quite a lot across contexts would be an interesting ablation study. - As LCE-A outperforms SAC quite consistently, LCE-A seems to the model of the choice. In contrast to SAC, including parameters of the embedding layer, the number of parameters in LCE-A can be larger than others. However, the complexity of LCE-A is not discussed.

Correctness: - Both SAC kernel and LCE-M kernel are derived simply by computing covariance between functions with certain independence assumptions. All these follows from basic kernel construction and correctly derived.

Clarity: 1. The main idea is simple and clear. The paper delivers the idea and details well enough to make it easy to follow. 2. Typo Eq(3) : 'k^x(x,x')' -> 'k^x(x_c,x'_c')' ??

Relation to Prior Work: Since the problem considered in this paper (aggregated rewards) has not been actively addressed, the difference made in the paper is quite clear.

Reproducibility: Yes

Additional Feedback:


Review 2

Summary and Contributions: [updated after rebuttal] I have read the rebuttal and other reviews. As also pointed out my other reviewers, the technical novelty is somewhat limited. So I'm not increasing my score. However, I think this paper is more of a practical contribution, I'm convinced the experiments are robust and it can be applied in a real industrial setting. Therefore, I still vote for acceptance. ------------------------------------------------------------------------------------------------ this paper introduces a new problem, high-dim contexture policy search with unknown context-level rewards, and proposes to solve it using Bayesian optimization with advanced modeling techniques. The target function is a reward function, mapping from policy parameters to reward. Under different context, the optimal policy parameters are different. However, context-level rewards can not be observed. Instead, only aggreated reward can be observed resulting from a distribution of contexts. So we must perform aggregate optimization where the parameters of all contexts must be optimized jointly, hence its dimension num_context by num_parameters is high. This work proposed two types of kernels, structural additive contextual (SAC) kernel and latent context embedding additive kernel (LCE-A). SAC assumes independence of the contexts and simply add up the kernels for each context weighted by the frequencies of the contexts. LCE-A learns an embedding of each context, then apply a kernel among contexts, and the context kernel and parameter kernel are multiplied and aggregated over all pairs of contexts. LCE-A is able to model the correlations among contexts. Two types of experiments are conducted. One is on synthetic function Hartmann6, where 5 dimensions are treated as parameters, and 1 dimention is treated as context, discretized uniformly. The experimental results are reasonable, observing context-level reward and optimize the policy independently is much better than only observing aggregate rewards. When we can't observe context-level reward, the proposed methods are better than treating the aggregate reward function as a total black-box. They also conduct a more "real" experiment optimizing adaptive video playback policy, the proposed methods are compared with a wide range of existing high-dim BO methods and LCE-A demonstrate the best performance.

Strengths: a novel and interesting problem setting that can be applied in practical industrial A/B test problems.

Weaknesses: there is not much technical novelty.

Correctness: appears to be correct

Clarity: the paper is well-written

Relation to Prior Work: I would put this work in the context of a recent research topic known as grey-box BO, and discuss the relationship.

Reproducibility: Yes

Additional Feedback: the authors claim that the proposed methods work well for a variety of contextual policy optimization use-cases at a large internet firm, would it be possible to describe one or two examples in some details without breaking anonymity and confidentiality (if any)? E.g. what's the problem setting, what performance is achieved compared to what baseline, etc. What are the embedding sizes for the LCE-A kernel? One thing I'm worried about is how did you learn a good embedding with only a few data points? The authors mentioned unsupervised pretraining in future work, which means there is no pre-train in this work. How do you explain the abnormally large variance of two data points (C0 and C4) in Figure 6 (right). one typo: L285: "10 %" -> "10%"


Review 3

Summary and Contributions: This paper presents a couple of kernels to deal with contextual BO with an application of adaptive bitrate policy for video playback.

Strengths: The empirical evaluation is strong. Although the numerical example is limited to a single function, and therefore, biased, the analysis is thoroughtly performed. The methods are also evaluated on a real problem, albeit simulated. The evaluation includes a comparison of many alternatives in the literature.

Weaknesses: The theoretical contribution is minimal. LCE-M kernel is also a special case of a SoS kernel, with known embeddings. In fact, in can be seen as a reformulation of the Kcc kernel of ICM. The SAC kernel seem to be a variant of the integrated response methods, that has been previously applied in BO. See reference below.

Correctness: The empirical evaluation is correct and the claims are reasonable.

Clarity: The paper is well written and easy to follow

Relation to Prior Work: Overall, the review of the state of art and related work is very thorough. However, I found the approach in section 3 very similar to the integrated responce methods in the EGO community. See for example and references therein: Tesch M, Schneider J, Choset H. Adapting control policies for expensive systems to changing environments. In2011 IEEE/RSJ International Conference on Intelligent Robots and Systems 2011 Sep 25 (pp. 357-364). IEEE.

Reproducibility: Yes

Additional Feedback: Update post-rebuttal: As commented before, the operation research community has studied this problem before. I now understand the differences with Tesch et al. but the aggregated/averaged/integrated response has also been addressed before. For example in the references [20, 21] in Tesch et al. assume that you cannot observe the environment and that it is randomly selected. You can also find interesting the thesis of Brian J. Williams. While the strategy presented here, specially for the LCE kernels, seems different, previous work should be addressed for a strong contribution. For the presented application/motivation, I recommend the authors also to clarify why this strategy is better than a two step process, where you first measure the environment (network quality) and then, you solve for that context. As far as I know as an outsider, the latter is closer to current methods for ABR. -------- The proposed method only affects the surrogate model via the kernel function. However, previous approaches on contextual/integrated approached had to also modify the acquisition function. For example, in Tesch et al, they found that the system focused on policies that performed well in "easy" contexts, but suboptimal in "difficult" context. Have this appeared in the experiments? Are context different enough to have easy/difficult contexts? Are the embeddings E() known in advance for the experiments? Can they be learned or adapted? Would that be similar to learning the whole B matrix as in [6]?


Review 4

Summary and Contributions: In this work authors study contextual policy optimization where the context rewards are not known. Gaussian process is used to model the reward. In contextual setting each context is treated as task. So multi-task GP is used to model reward in multiple contextual settings. The modeling of the covariance of the MTGP is done with context-parameter separable kernels . The authors explains their method with elaborate experiments.

Strengths: 1.The paper introduces the contextual policy optimization with unknown context reward. 2. The paper focuses on real -world problem (adaptive bit rate) in contextual policy optimization setting. 3. The paper exploits reward -context separability to improve over the existing model.

Weaknesses: Please see the comments.

Correctness: The paper looks correct.

Clarity: The paper is mostly well written. A section with problem statement would be helpful.

Relation to Prior Work: The paper clearly states the relevant prior work in Bayesian optimization, Multi-task case and Contextual case.

Reproducibility: Yes

Additional Feedback: 1. The paper needs a problem statement section clearly explaining the problem and goal. 2. The concise description of the section 2 and 3 would be better. 3. In Section 2.1 for ICM , the paper states that matrix B is learned. What is the learning method and how well can it learned? 4. In section 2.2 for LCM, the dimension of the problem increases. As stated earlier that BO struggles with high dimension (>15). Then why LCM is used ? ========================================================= The authors have performed detailed experiments but I have found the theoretical contribution to be limited. This is not my area of work. So I am unsure of the impact of this work in this field. I am keeping my score the same.

[Author Response · NeurIPS 2020]

We thank the reviewers for their critical assessment of our work, and for the overall positive reception. Below we reply
to each review to clarify points and provide additional insights into the problem and methods, which we hope will better
highlight the technical novelty and contribution of our work.

**Reviewer 1**

• To the first point in weaknesses. We agree on this important point and we use cross-validation to verify this
assumption. For instance, in our real-world applications (described below for R2), the model fit of LCE-A is much
better than other models. Note also that the ABR simulator uses real throughput traces and video data, and the results
there indicate a robustness of assuming a shared spatial kernel.

• To the second point. Thanks for raising this! The number of parameters is $(1 + m) \times C + d + m$ for embedding
dimension $m$. Although LCE-A has more parameters, the model complexity is not necessarily higher. The kernel
$k^z(E(c), E(c'))$ imposes regularization in the model and encourages correlations across component functions. Lines
124–128 in Sec 2.2 discuss the connection between the regularization of a GP prior and the function norm in the
RKHS. With stronger correlation there is a larger norm and more regularization. We will clarify this in the paper.

**Reviewer 2**

• To the first point in comments. Thanks for asking! We have applied this method to two real problems: 1) tuning a
live video playback controller under different connection qualities, to minimize stall time while maintaining high
quality. There are 6 controller parameters and 5 connection qualities (from excellent to poor). 2) tuning data fetch
parameters for an app surface, contextualized for different countries and connection types (4g, wifi, etc). The goal
was to reduce data utilization without affecting app performance. For both problems, performance metrics had to be
evaluated by A/B tests that took several days (though multiple design points could be run in parallel). Metrics could
not be logged at the level of context, meaning typical contextual BO methods could not be used. For both problems,
the LCE-A model had the best cross-validation performance and so was used for the optimization. LCE-A found the
best policy compared with random search and standard BO, and significantly improved system performance.

• To the second point. The embedding sizes are set to be 1 in the simulation studies. If LCE-A fails to learn the
embedding and the corresponding correlation structure, then it essentially falls back to SAC, which doesn't borrow
strength across contexts; we will add discussion of this. As noted, when possible, pre-training can accelerate learning
in the small-data regime, and is something we will investigate in future work.

• The two context buckets have very different network conditions and their bandwidths are unstable; thus the QoEs
have larger variances. We will clarify this in the paper.

• Thanks for the suggestion in related work! We will add this into the discussion. Our work assumes an additive
structure of component functions and fits nicely into the framework of grey-box BO $g(h(\mathbf{x}))$. Here, $g$ is an additive
function and $h$ consists of unknown context functions which are learnt via the new LCE kernel.

**Reviewer 3**

• Thanks for the reference [TSC11]! We were unaware of this paper, but it is indeed related and our work nicely
extends the methods developed there. Eqn. (1) of TSC11 is the problem that we solve as well. When using the
LCE-A and SAC kernel, the acquisition function was the same as their expected policy score improvement. Our work
extends TSC11 by handling the situation where the outcomes $f(x_e, x)$ (i.e. $f_c(x_c)$ in our paper) are not observable
under each environment (i.e. context) separately. We show how to optimize Eqn. (1) of TSC11 when given only
total score, which is critical for settings where the controller operates across heterogeneous environments and score
cannot be easily attributed to each (the notorious credit assignment problem). We will add discussion of TSC11 and
its area of work, and we believe that the connection to it broadens and strengthens the technical contribution of our
paper. We hope that clarifying this point has made the contribution of our work more apparent.

• On policy performance in easy/difficult contexts. In our simulation studies, the easy contexts are the dense contexts
with the majority of the weight and the difficult contexts are the sparse contexts with small weights. We show that
LCE-A improves performance in difficult contexts by learning the context correlation structure, e.g. in Fig. 4(b).

• To the second question in comments. In our current studies, embeddings are learnt similar to inferring the matrix $B$
inside an ICM kernel by maximizing likelihood. We will clarify this.

**Reviewer 11**

• Thanks for your suggestions on improving organization of the paper! For Q3, the matrix $B$ is estimated by maximizing
marginal likelihood, as is typical for multi-task GP fitting.

• For Q4, the struggle BO has with high dimensions is with the number of parameters in the feature space. That is
remaining constant across these models. LCE-M does have additional hyperparameters, but these tend to increase
regularization and do not cause problems of the sort faced by high-dimensional BO (see response to R1).

[Meta-Review · NeurIPS 2020]

This paper considers an interesting problem setting where reward is biased by context. It provides a strong and thorough experimental analysis on real-world problems that includes a comparison to several competing techniques, and the results are compelling. On the negative side, the approach lies mainly on existing techniques and the level of conceptual advance and innovation is rather low, the presentation should be improved to provide a clearer problem statement, and the relationship to prior related work in the field of Operations Research is missing. On balance, the consensus opinion of the reviewers is that the problem and experimental analysis makes a sufficient conversation and that the authors are committed to addressing the improvements mentioned above in their final version. The recommendation is to accept on this presumption.